# Nitric oxide-dependent anaerobic ammonium oxidation

Ziye Hu[1,3], Hans J.C.T. Wessels[2], Theo van Alen[1], Mike S.M. Jetten[1] & Boran Kartal[1,4]

Nitric oxide (NO) has important functions in biology and atmospheric chemistry as a toxin, signaling molecule, ozone depleting agent and the precursor of the greenhouse gas nitrous oxide ($N_2O$). Although NO is a potent oxidant, and was available on Earth earlier than oxygen, it is unclear whether NO can be used by microorganisms for growth. Anaerobic ammonium-oxidizing (anammox) bacteria couple nitrite reduction to ammonium oxidation with NO and hydrazine as intermediates, and produce $N_2$ and nitrate. Here, we show that the anammox bacterium *Kuenenia stuttgartiensis* is able to grow in the absence of nitrite by coupling ammonium oxidation to NO reduction, and produce only $N_2$. Under these growth conditions, the transcription of proteins necessary for NO generation is downregulated. Our work has potential implications in the control of $N_2O$ and NO emissions from natural and manmade ecosystems, where anammox bacteria contribute significantly to $N_2$ release to the atmosphere. We hypothesize that microbial NO-dependent ammonium oxidation may have existed on early Earth.

[1] Department of Microbiology, IWWR, Radboud University Nijmegen, Heyendaalseweg 135, 6525AJ Nijmegen, The Netherlands. [2] Translational Metabolic Laboratory, Department of Laboratory Medicine, Radboud University Medical Center, Geert Grooteplein-zuid 10, 6525GA Nijmegen, The Netherlands. [3] Present address: Sanquin, Plesmanlaan 125, 1066 CX Amsterdam, The Netherlands. [4] Present address: Microbial Physiology Group, Max Planck Institute for Marine Microbiology, Celsiusstraße 1, 28359 Bremen, Germany. Correspondence and requests for materials should be addressed to B.K. (email: bkartal@mpi-bremen.de)

Nitric oxide (NO) is a highly reactive molecule that plays a key role in atmospheric chemistry as an ozone depleting agent and as the precursor of the greenhouse gas nitrous oxide ($N_2O$)[1–3]. In biology NO has several distinct functions: it is a very potent toxin, but also a signaling molecule and an intermediate in the microbial nitrogen cycle[4]. To remove toxic NO, microorganisms use a multitude of enzymes, and intricate mechanisms that sense, scavenge, and convert NO to less reactive compounds such as $N_2O$[5]. Due to its high reactivity, NO exists only at very low concentrations in cells, and is rapidly turned over, which is why in both denitrification and aerobic ammonia oxidation it was the last recognized intermediate[6–8].

In microbial processes, NO is either generated via nitrite reduction catalyzed by Cu-containing (Cu-NIR) or cytochrome $cd_1$-containing ($cd_1$-NIR) nitrite reductases[7,9], or alternatively by hydroxylamine oxidation catalyzed by octaheme hydroxylamine oxidoreductases (HAO)[6,10]. Intriguingly, microbial growth with NO as the terminal electron acceptor have not been shown before. However, before molecular oxygen started to accumulate in the atmosphere, NO was the strongest oxidant available on earth[11–14], which suggests that microorganisms capable of using external NO as the terminal electron acceptor could have evolved early in the history of life.

Anaerobic ammonium-oxidizing (anammox) bacteria use the oxidative power of NO to activate ammonium in the absence of oxygen. These microorganisms normally use nitrite as their terminal electron acceptor, producing nitrate and $N_2$[15]. Their catabolism can be described in three main reactions: First, $NO_2^-$ is reduced to NO (Eq. (1)); then NO and $NH_4^+$ are condensed into hydrazine ($N_2H_4$) by hydrazine synthase (Eq. (2)), which is followed by the oxidation of $N_2H_4$ to $N_2$ by hydrazine dehydrogenase (Eq. (3)). The four electrons released from hydrazine oxidation are used for nitrite reduction (1 electron) and hydrazine synthesis (3 electrons) completing the anammox catabolic cycle. The electrons that are necessary for cell carbon fixation are suggested to be delivered by the oxidation of nitrite to nitrate (Eq. (4)) based on the observation that the growth of anammox bacteria appears to be always associated with nitrate production[15,16].

$$NO_2^- + 2H^+ + e^- \rightarrow NO + H_2O (E_0^{'} = +0.38\,V) \quad (1)$$

$$NO + NH_4^+ + 2H^+ + 3e^- \rightarrow N_2H_4 + H_2O (E_0^{'} = +0.06\,V) \quad (2)$$

$$N_2H_4 \rightarrow N_2 + 4H^+ + 4e^- (E_0^{'} = -0.75\,V) \quad (3)$$

$$NO_2^- \rightarrow NO_3^- + 2H^+ + 2e^- (E_0^{'} = +0.42\,V) \quad (4)$$

While the enzymes responsible for reactions (2)–(4) are conserved in all known anammox genera[17], nitrite reduction to NO is catalyzed by distinct enzymes that are also found in other nitrogen-transforming microorganisms[7,9]. The anammox species *Kuenenia stuttgartiensis* and *Scalindua profunda* encode $cd_1$-NIR[18–20], *Jettenia* spp. encode Cu-NIR[21], whereas *Brocadia* spp. do not encode any known nitrite reductases[22]. Moreover, all anammox bacteria encode an octaheme HAO that catalyzes the oxidation of hydroxylamine to NO[10,17]. It is evident that different anammox species have different NO-forming pathways, which suggests that nitrite reduction to NO might be a trait that was acquired after the core anammox catabolism was already in place. Indeed, reactions (2) and (3) would be sufficient both to conserve energy and supply necessary electrons for cell carbon fixation ($CO_2$) for biomass. In this scenario, three of the four electrons released from hydrazine oxidation would be used for hydrazine synthesis, and the remaining electron could be used for biomass production, without the need for nitrite oxidation to nitrate.

To test this hypothesis, a free-living planktonic *K. stuttgartiensis* culture continuously supplied with ammonium and NO as the only substrates in a continuous membrane bioreactor is employed. We show that *K. stuttgartiensis* is able to use NO as its terminal electron acceptor, and conserve energy and grow by coupling NO reduction to ammonium oxidation in the absence of nitrite. Under these conditions, nitrate is not produced and the sole end product is $N_2$. Using comparative transcriptomics and proteomics, we demonstrate that when growing on NO-dependent ammonium oxidation, *K. stuttgartiensis* down regulates the transcription of proteins responsible for NO generation as well as nitrite oxidation.

## Results

**NO-dependent anaerobic ammonium oxidation.** All continuous bioreactors were operated with free-living planktonic *K. stuttgartiensis* cell cultures (more than 95% enriched) for more than 50 days (~5 generations). Continuous bioreactors have an intrinsic reproducibility of the measured growth rate, which can be established by removing biomass at a constant rate, and measuring whether the newly grown cells exhibit the same activity for extended periods of time. Here, biomass was constantly removed with a rate of 120 ml day$^{-1}$ from each bioreactor, all of which had stable activity and growth rate (Fig. 1). Within the first week after inoculation, NO was introduced to reactors II and III and the NO concentration was increased to 450 mg-N l$^{-1}$ (32 mM).

In reactor II, where NO was supplied as an additional substrate next to ammonium and nitrite, NO consumption was accompanied with an increase in ammonium oxidation, in line with an earlier study that used flocculent biomass[24]. Ammonium concentration in the effluent decreased from 100 mg-N l$^{-1}$ (7 mM) to 25 mg-N l$^{-1}$ (1.8 mM), and stayed at this level for the rest of the reactor operation indicating that the anammox bacteria oxidized 228 mg-N l$^{-1}$ (16.3 mM) ammonium coupled to NO reduction (Fig. 1b). In this reactor, NO consumption was $359 \pm 6$ mg-N l$^{-1}$ day$^{-1}$ (Fig. 1b, average of last 40 days, including standard deviation), corresponding to ~80% of the NO load.

Reactor III only received ammonium and NO as substrates. Here, 120 mg-N day$^{-1}$ (8.6 mmol day$^{-1}$) ammonium was oxidized coupled to reduction of 318 mg-N l$^{-1}$ NO (Fig. 1c, average of last 44 days: 70% of the NO supplied). This activity was accompanied with a stable growth rate and the ratio of reduced NO to oxidized ammonium was 1.59, which was close to the predicted stoichiometry of 1.5 (Eq. (5)). Taken together, these results clearly showed that the anammox bacteria were able to conserve energy and grow from anaerobic ammonium oxidation coupled to NO reduction in the absence of nitrite.

$$6NO + 4NH_4^+ \rightarrow 5N_2 + 6H_2O + 4H^+ (\Delta G_0^{'} = -1784) \quad (5)$$

**Anammox bacteria do not detoxify NO to $N_2O$.** Besides hydrazine synthase, which uses NO to activate ammonium, anammox bacteria also encode NO-reducing enzymes, such as the flavoprotein norVW (kuste3160 in *K. stuttgartiensis*). Therefore, these microorganisms have the genetic potential to detoxify NO to $N_2O$. Still, under all growth conditions, the formed $N_2O$ was only a minor fraction of the nitrogen load, ~0.025%, ~0.04%, ~0.09% in reactors I, II, and III, respectively. Even though reactors II and III were fed high concentrations of NO, only 0.12% (Reactor II) and 0.18% (Reactor III) of removed NO was converted to $N_2O$. In line with this observation, transcription levels of norVW (kuste3160) were just above detection

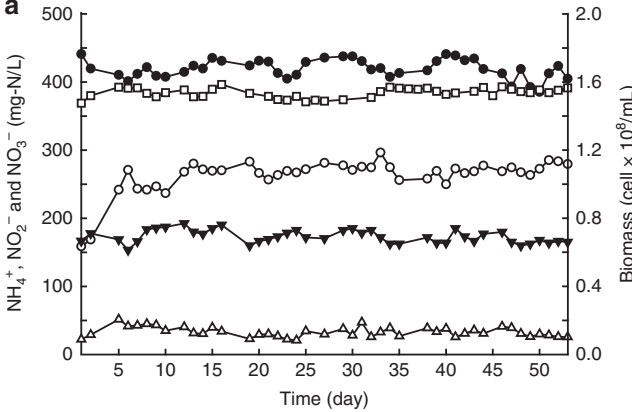

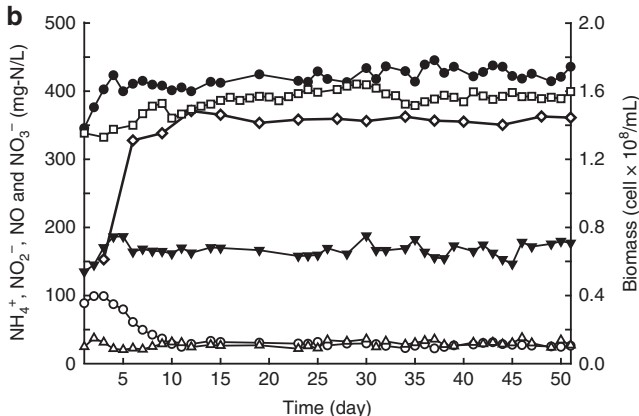

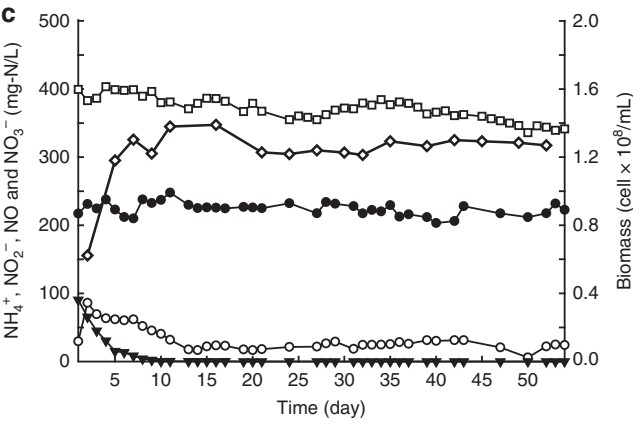

**Fig. 1** Nitric oxide, nitrite, ammonium consumption, nitrate production, and biomass growth during reactor operation. (**a**) reactor I (control reactor, supplied with ammonium and nitrite), (**b**) reactor II (supplied with ammonium, nitrite, and NO) and (**c**) reactor III (supplied with ammonium and NO). Filled and empty circles indicate ammonium concentration in the influent and effluent, respectively. Empty triangles indicate nitrite concentration in the influent. Nitrite concentration in the effluent was always below detection limit. Filled triangles indicate nitrate concentration in the effluent. Cell concentration is displayed by cell numbers per ml (open squares). Nitric oxide (empty diamonds) is displayed as consumption by the bioreactors. This is calcualted from the difference between concentration of nitric oxide in the influent and effluent of the gas phase of the bioreactor. The source data underlying this figure are provided as Source Data file

limit (~10 reads), and did not change in the reactors that were supplied with NO. The small amount of $N_2O$ production was likely due to the other community members (<5%) present in the *K. stuttgartiensis* cultures, which might be reducing NO to $N_2O$.

using minute amounts of organic carbon originating from cell decay[19].

**Nitrite oxidation is not needed for growth of anammox bacteria**. Anammox growth on ammonium and nitrite is always accompanied by nitrate production, which accounts for up to 20% of the consumed nitrite[16,25]. Oxidation of nitrite to nitrate is suggested to supply the electrons needed for cell carbon ($CO_2$) fixation[26]. Indeed, nitrate production was observed in both reactor I that was supplied ammonium and nitrite, and reactor II, which received ammonium, nitrite, and NO (Fig. 1a, b). The average nitrate concentration in the effluents of reactors I and II (average of last 40 days) were very similar, 29 mg-N l[−1] (~2.1 mM), and 32 mg-N l[−1] (~2.3 mM), respectively.

Conversely, in reactor III, which was supplied with ammonium and NO, nitrate concentration in the effluent decreased to below detection limit (0.1 μM) within 1 week after the start of the NO supply, and was not detected for the remainder of the experimental period (Fig. 1c). Apparently under these growth conditions, nitrate was not produced, indicating that anammox bacteria do not acquire the electrons for cell carbon fixation from the oxidation of nitrite to nitrate.

This observation resolves the thermodynamic challenge of reverse electron transport that arises if high potential electrons released from nitrite oxidation to nitrate ($E_0' = +0.43$ V) would be used for carbon fixation via the acetyl-CoA pathway, which includes several reactions with very low redox potential (e.g. $E_0' = -0.42$ V for the reduction of $CO_2$ to formate). On the other hand, coupling the low potential electrons released from hydrazine oxidation to $N_2$ ($E_0' = -0.75$ V) to carbon fixation would be very efficient. Indeed, when growing as NO-dependent ammonium oxidizers, this is what the anammox bacteria do.

$$3NO_2^- + 2H^+ \rightarrow 2NO + NO_3^- + H_2O \qquad (6)$$

It is highly likely that also when anammox bacteria grow on nitrite and ammonium, the low potential electrons released from hydrazine oxidation to $N_2$ are directed to cell carbon fixation, instead of using electrons from nitrite oxidation to nitrate for this purpose. Consequently, nitrite oxidation to nitrate ($E_0' = +0.43$ V) and nitrite reduction to NO ($E_0' = +0.38$ V) are most likely coupled to one another (Eq. (6)). This is thermodynamically more favorable than coupling nitrite oxidation to cell carbon fixation, and indicate that anammox bacteria essentially disproportionate nitrite to nitrate and NO (Eq. (6)).

**Pathways related to nitrite turnover are down-regulated**. The use of NO as the sole electron acceptor was also reflected in the transcriptional activity of the anammox bacteria. When growing on NO and ammonium, 136 genes were up-regulated, and 252 were down regulated over five-fold (Supplementary Data 1 and Supplementary Data 2). As nitrite was not supplied, the most abundant nitrite transporter (kuste3055) was down-regulated 36-fold. In line with earlier reports, transcripts of genes encoding the catalytic subunit of the cytochrome $cd_1$ nitrite reductase (nirS, kuste4136) and its accessory proteins (kuste4137–kuste4140) were detected in low amounts even under normal nitrite-dependent ammonium-oxidizing conditions[18]. Nevertheless, under NO-dependent ammonium-oxidizing conditions, all genes in this cluster were strongly down regulated (between 15 and 113 fold): transcription of nirS was 52 fold lower, while the transcription of its proposed redox partner, nirC (kuste4137) decreased from 71 RPKM in the control reactor to below the detection limit (0 reads) (Table 1).

Based on the fact that nirS was not abundant in the transcriptome and proteome analyses of *K. stuttgartiensis*, it

**Table 1 Down-regulation of *Kuenenia stuttgartiensis* genes involved in nitrite and NO metabolisms**

| ORF | Gene | Annotation | Unique mapped reads | | Down-regulation factor | |
|---|---|---|---|---|---|---|
| | | | Reactor II | Reactor III | Reactor II | Reactor III |
| *Hydroxylamine oxidoreductase-like and associated proteins* | | | | | | |
| kustc0457 | | Diheme cytochrome *c* protein | 181 | 8 | 1.6 | 39.1 |
| kustc0458 | hao | Similar to hydroxylamine oxidoreductase (hao) | 476 | 32 | 1.7 | 27.8 |
| kustc1061 | hox | Hydroxylamine:nitric oxide oxidoreductase | 1949 | 384 | 1.3 | 7.2 |
| kuste4574 | hao | Similar to hydroxylamine oxidoreductase (hao) | 147 | 122 | 1.8 | 2.6 |
| *Nitrite:nitrate oxidoreductase gene cluster* | | | | | | |
| kustd1699 | | Hypothetical protein | 105 | 15 | 2.3 | 18.2 |
| kustd1700 | NxrA | Nitrite:nitrate oxidoreductase reductase subunit A | 1296 | 326 | 2.1 | 9.3 |
| kustd1701 | | Unknown protein | 168 | 82 | 2.4 | 5.5 |
| kustd1702 | | Hypothetical protein | 73 | 18 | 4.2 | 18.9 |
| kustd1703 | NxrB | Nitrite:nitrate oxidoreductase reductase subunit B | 334 | 159 | 2.1 | 4.9 |
| kustd1704 | | Unknown protein | 409 | 103 | 1.9 | 8.2 |
| kustd1705 | c556 | Conserved hypothetical (monoheme) protein | 30 | 7 | 2.5 | 11.7 |
| kustd1706 | uspA | Similar to universal stress protein UspA | 42 | 24 | 1.2 | 2.4 |
| kustd1707 | | Conserved hypothetical (monoheme) protein | 17 | 7 | 1.3 | 3.5 |
| kustd1708 | | Hypothetical (tetraheme) protein | 74 | 35 | 1.2 | 2.8 |
| kustd1709 | | Hypothetical (diheme) protein | 102 | 61 | 1.2 | 2.2 |
| kustd1710 | | Hypothetical (monoheme) protein | 35 | 15 | 1.9 | 5.0 |
| kustd1711 | | Hypothetical (diheme) protein | 63 | 22 | 1.5 | 4.7 |
| kustd1712 | cydA | Similar to subunit 1 of alternative cytochrome *bd* quinol oxidase | 86 | 32 | 1.7 | 5.2 |
| kustd1713 | | Conserved hypothetical protein | 79 | 15 | 1.3 | 7.7 |
| *Nitrite transporter gene cluster* | | | | | | |
| kuste3050 | | Putative nitrite transporter | 14 | 6 | 2.9 | 7.5 |
| kuste3051 | | Unknown protein | 7 | 2 | 2.6 | 10.0 |
| kuste3052 | | Hypothetical protein | 119 | 115 | 2.6 | 3.0 |
| kuste3053 | | Hypothetical protein | 118 | 93 | 2.2 | 3.1 |
| kuste3054 | | Unknown protein | 94 | 44 | 2.1 | 5.0 |
| kuste3055 | | Conserved hypothetical protein | 250 | 25 | 3.3 | 36.5 |
| *Nitrite reductase gene cluster* | | | | | | |
| kuste4136 | nirS | Strongly similar to *cd*$_1$ nitrite reductase NirS | 86 | 5 | 2.7 | 51.8 |
| kuste4137 | nirC | Hypothetical mono heme cytochrome NirC | 10 | 0 | 3.0 | 27.1 |
| kuste4138 | nirN | Conserved hypothetical NirN maturation protein | 32 | 1 | 3.2 | 112.7 |
| kuste4139 | nirJ | Similar to NirJ/MoaA/PpqE family of cofactor | 16 | 1 | 2.4 | 41.8 |
| kuste4140 | nirF | Similar to heme *d*$_1$ biosynthesis protein NirF | 14 | 3 | 2.9 | 14.9 |

Reactor II (fed with nitrite, NO and ammonium) and Reactor III (fed with NO and ammonium) are compared to Reactor I (fed with nitrite and ammonium)

was suggested that other proteins could (also) be responsible for nitrite reduction to NO. Comparative sequence analyses had indicated that octaheme hydroxylamine oxidoreductase (HAO)-like proteins encoded by kustc0458 and kuste4574, which have homologs in all known anammox genera, could act as nitrite:NO oxidoreductases[15,17]. The transcriptome analyses showed that octaheme kustc0458 and its diheme redox partner kustc0457 were down-regulated 39 and 28 fold, respectively, in the NO-fed reactor, and were among the 15 most down-regulated proteins. Kuste4574 was also down regulated, albeit only 2.6 fold. Furthermore, kustc1061, which is a dedicated octaheme HAO that oxidizes hydroxylamine to NO, was 7 fold down regulated (Table 1). Taken together, our results suggested that these HAO-like proteins were involved in in vivo NO production, and kustc0458 could be one of the main enzymes responsible for nitrite reduction to NO in *K. stuttgartiensis*. In line with the observation that nitrate production did not occur in the NO-fed reactor, the entire gene cluster (kustd1699–kustd1713) that contains nitrite:nitrate oxidoreductase (NXR) and its accessory proteins was down regulated. The genes that encode the soluble, catalytic component, composed of nxrA (kustd1700), nxrB (kustd1703), and nxrC (kustd1704), were down regulated 9.3, 4.9 and 8.2 fold, respectively (Table 1).

The down regulation in the transcription of genes involved in nitrite turnover was not reflected at the protein level (Table 2). The vast majority of proteins did not show a dramatic change in abundance between culturing conditions. This discrepancy between transcript and protein levels is a common observation in studies that combine transcriptomics and proteomics since individual protein levels are determined by many different factors that govern the balance between protein synthesis and protein turnover[27]. It is therefore important to integrate rather than correlate both data types to derive novel biological insights[28]. Apparently, anammox bacteria do not rapidly degrade the majority of their proteins. While fast dividing organisms require much higher protein synthesis (and degradation) rates to prevent protein dilution while dividing, the strategy not to rapidly degrade and resynthesize proteins makes sense for slow-growing microorganisms. In this way, they would be able to keep their protein complement primed for conditions that might occur, and thereby reduce their response time to changing substrates.

## Discussion

In the present study, we showed that in the absence of nitrite, anammox bacteria couple ammonium oxidation stoichiometrically to NO reduction, and produce N$_2$ as the sole end product. Taken together with earlier results[24], our results indicated that this was a common trait in all anammox bacteria. Such a metabolism could have existed on ammonium-rich early earth, where NO was present before the advent of more oxidized nitrogen species such as nitrite and nitrate. Indeed,

**Table 2 Detection of proteins involved in nitrite and NO metabolisms**

| ORF | Gene name | Annotation | Label-free quantification values | | | Log2 ratio | |
|---|---|---|---|---|---|---|---|
| | | | Reactor I | Reactor II | Reactor III | II/I | III/I |
| *Nitrite transporter gene cluster* | | | | | | | |
| kuste3050 | | Putative nitrite transporter | – | – | – | – | – |
| kuste3051 | | Unknown protein | – | – | – | – | – |
| kuste3052 | | Hypothetical protein | 1,158,000 | 336,465 | 1,202,850 | −1.78 | 1.84 |
| kuste3053 | | Hypothetical protein | – | – | – | – | – |
| kuste3054 | | Unknown protein | 335,650 | 196,738 | 267,740 | −0.77 | −0.33 |
| kuste3055 | | Conserved hypothetical protein | 471,695 | 195,227 | 380,145 | −1.2727 | 0.9614 |
| *Nitrite reductase gene cluster* | | | | | | | |
| kuste4136 | nirS | Strongly similar to $cd_1$ nitrite reductase NirS | 888,748 | 1,303,250 | 952,540 | 0.55 | −0.45 |
| kuste4137 | nirC | Hypothetical mono heme cytochrome NirC | – | – | – | – | – |
| kuste4138 | nirN | Conserved hypothetical NirN maturation protein | – | – | – | – | – |
| kuste4139 | nirJ | Similar to NirJ/MoaA/PpqE family of cofactor | – | – | – | – | – |
| kuste4140 | nirF | Similar to heme $d_1$ biosynthesis protein NirF | – | – | – | – | – |
| *Hydroxylamine oxidoreductase-like and associated proteins* | | | | | | | |
| kustc0457 | | Diheme cytochrome *c* protein | 1,341,000 | 2,005,250 | 904,210 | 0.58 | −1.15 |
| kustc0458 | hao | Similar to hydroxylamine oxidoreductase (hao) | 2,266,300 | 3,095,925 | 1,553,175 | 0.45 | −1.00 |
| kustc0694 | hdh | Hydrazine dehydrogenase | 10,175,950 | 12,669,000 | 6,790,225 | 0.32 | −0.90 |
| kustc1061 | hox | Hydroxylamine:nitric oxide oxidoreductase | 6,259,425 | 9,912,625 | 4,616,800 | 0.66 | −1.10 |
| *Hydrazine synthase gene cluster* | | | | | | | |
| kuste2854 | | Hypothetical (triheme) protein | 411,257 | 499,910 | 263,705 | 0.28 | −0.92 |
| kuste2855 | | Hypothetical (hepta heme) protein | – | – | – | – | – |
| kuste2856 | fdoI | Similar to formate dehydrogenase, cytochrome $b_{556}$ subunit | 216,773 | 93,117 | 238,043 | −1.22 | 1.35 |
| kuste2857 | hydG/atoC | Strongly similar to sigma 54 response regulator | – | – | – | – | – |
| kuste2858 | | Unknown protein | 144,710 | 178,908 | – | 0.31 | – |
| kuste2859 | hzsC | Hydrazine synthase subunit C | 18,435,000 | 25,679,250 | 22,060,750 | 0.48 | −0.22 |
| kuste2860 | hzsB | Hydrazine synthase subunit B | 16,229,500 | 19,931,250 | 15,917,250 | 0.30 | −0.32 |
| kuste2861 | hzsA | Hydrazine synthase subunit A | 26,991,750 | 36,862,250 | 23,070,000 | 0.45 | −0.68 |
| *Nitrite:nitrate oxidoreductase gene cluster* | | | | | | | |
| kustd1699 | | Hypothetical protein | 7,217,425 | 9,114,050 | 7,424,300 | 0.34 | −0.30 |
| kustd1700 | nxrA | Nitrite:nitrate oxidoreductase reductase subunit A | 11,748,750 | 18,509,500 | 11,701,750 | 0.66 | −0.66 |
| kustd1701 | | Unknown protein | – | – | – | – | – |
| kustd1702 | | Hypothetical protein | – | – | – | – | – |
| kustd1703 | nxrB | Nitrite:nitrate oxidoreductase reductase subunit B | 4,353,575 | 6,927,575 | 3,811,600 | 0.67 | −0.86 |
| kustd1704 | | Unknown protein | 4,152,800 | 6,798,625 | 4,091,800 | 0.71 | −0.73 |
| kustd1705 | c556 | Conserved hypothetical (monoheme) protein | 1,131,375 | 1,773,775 | 733,265 | 0.65 | −1.27 |
| kustd1706 | uspA | Similar to universal stress protein UspA | – | – | – | – | – |
| kustd1707 | | Conserved hypothetical (monoheme) protein | – | 81,870 | – | NaN | – |
| kustd1708 | | Hypothetical (tetraheme) protein | 675,875 | 650,110 | 494,425 | −0.06 | −0.39 |
| kustd1709 | | Hypothetical (diheme) protein | 426,010 | 287,390 | 481,298 | −0.57 | 0.74 |
| kustd1710 | | Hypothetical (monoheme) protein | – | – | – | – | – |
| kustd1711 | | Hypothetical (diheme) protein | – | – | 271,150 | – | – |
| kustd1712 | cydA | Similar to subunit 1 of alternative cytochrome *bd* quinol oxidase | – | – | – | – | – |
| kustd1713 | | Conserved hypothetical protein | 4,680,375 | 5,748,750 | 6,014,050 | 0.30 | 0.07 |

Reactor I is fed with nitrite and ammonium, reactor II is fed with nitrite, NO and ammonium and reactor III is fed with NO and ammonium

the variability of the nitrite-reducing enzymes in different anammox genera suggests that the ability to reduce nitrite to NO could have been acquired at a later stage. When growing with NO, anammox bacteria did not produce nitrate, refuting the longstanding assumption that nitrate production indicates the growth of anammox bacteria, and that nitrite oxidation to nitrate is required for cell carbon fixation. Instead, anammox bacteria most likely perform nitrite disproportionation by coupling nitrite reduction to NO with nitrite oxidation to nitrate. Indeed, other forms of disproportionation of nitrite (into $N_2$ and nitrate or $N_2O$ and nitrate) and NO (into $N_2$ and nitrate or $N_2O$ and nitrite or $N_2O$ and nitrate) might be a widespread trait of nitrogen-transforming microorganisms as both reactions are thermodynamically favorable, and can be carried out by already known enzymes[29]. When anammox bacteria were fed by NO and ammonium only, all proteins involved in nitrite uptake and consumption were transcriptionally down-regulated.

This observation also singled out the octaheme HAO protein encoded by kustc0458 as the most likely candidate enzyme that reduces nitrite to NO in *K. stuttgartiensis*. The anammox genus *Brocadia* lacks both Cu-NIR and $cd_1$-NIR, but contains kustc0458 homologs, which could be reducing nitrite to NO in *Brocadia*. The changes in the protein complement of *K. stuttgartiensis* cells were comparatively less prominent. This highlights the fact that transcriptional regulation is not always reflected by similar changes in protein levels, and that observation of a protein does not necessarily mean that the activity it is implicated in is taking place.

Finally, even though they were supplied large quantities of NO, anammox bacteria did not produce any $N_2O$, suggesting that in natural and manmade ecosystems these microorganisms consume NO, and convert it to harmless $N_2$ instead of the greenhouse gas $N_2O$. Consequently, they contribute to controlling the emissions of both NO and $N_2O$ two central molecules in atmospheric chemistry.

## Methods

**Reactor setup and operation**. Three continuous membrane bioreactors (working volume, 2 l) were inoculated with an equal amount (1 l, $OD_{600} = 0.8$) of highly enriched (more than 95% *K. stuttgartiensis* cells[18]) free-living planktonic cell suspensions of the anammox bacterium *K. stuttgartiensis,* and were operated simultaneously for more than 50 days. The reactors were flushed continuously with $Ar/CO_2$ (95%/5%, 10–15 ml min$^{-1}$) to maintain anaerobic conditions. The temperature and pH of the reactors were maintained at 30 °C and 7.3 with a water bath, and 1 M $KHCO_3$ solution, respectively. The reactors were stirred at 600 rpm with two impellors, one installed just above the gas inlet and the other below the gas–liquid interface of the reactors. Synthetic medium[16] was supplied at a flow rate of 500–600 ml day$^{-1}$ to all three reactors. Reactors I and II were supplied with 420 mg-N l$^{-1}$ (30 mM) ammonium and 168 mg-N l$^{-1}$ (12 mM) nitrite, whereas reactor III was supplied with 210 mg-N l$^{-1}$ (15 mM) ammonium and no nitrite. NO (10,000 ppm, in Argon) was introduced to reactors II and III 36 h after the startup with a flow rate of 10 ml min$^{-1}$ and increased to 30 ml min$^{-1}$ within 2–3 days. NO was bubbled through the reactor in gaseous form. Reactor I was operated as a control reactor and was not supplied with NO. NO consumption was calculated using the difference between the NO concentration in the influent gas phase and the effluent gas phase, taking the flow rate into consideration. The flow rate of $Ar/CO_2$ in reactor I was increased to 45 ml min$^{-1}$ to maintain the same total gas flow rate as the other two reactors. To assess growth, optical density at 600 nm ($OD_{600}$) was monitored during the whole experimental period, which was used to calculate cell numbers per milliliter of reactor volume. After inoculation, when $OD_{600}$ was stable, for each reactor, biomass was removed constantly from each bioreactor at a rate of 120 ml day$^{-1}$.

**Analytical methods**. Liquid samples (1 ml) were collected from the influent and effluent of each reactor every 2–3 days, and were pelleted by centrifugation for 5 min at $16,000 \times g$. The supernatants were stored at −20 °C until further analyses. Nitrite concentrations were determined colorimetrically at 540 nm after a 20 min reaction of 1 ml sample with 1 ml 1% sulfanilic acid in 1 M HCl and 1 ml 0.1% N-naphtylethylenediamine[23]. Ammonium concentrations were determined colorimetrically at 420 nm after a 30 min reaction of 40 μl sample with 760 μl 0.54% *ortho*-phthalaldehyde, 0.05% β-mercaptanol, and 10% ethanol in 400 mM potassium phosphate buffer (pH 7.3)[23]. The nitrate concentrations were determined by a Sievers Nitric Oxide Analyzer 280i (Analytix Ltd, UK) according to the manufacturer's instructions. The gas inlet and outlet of reactors II and III were connected to an Eco Physics CLD700 EL chemiluminescence $NO_x$ analyzer (EcoPhysics, Switzerland) every 3–4 days to measure the concentration of NO. $N_2O$ was measured at least once a week with an Agilent 6890 Series GC (Agilent Technologies, USA) equipped with a Porapak Q column and an electron capture detector (ECD).

**RNA isolation, transcriptome sequencing, and analyses**. Biomass (8 ml) was harvested from all three reactors on day 20 and pelleted by centrifugation for 5 min at $16,000 \times g$ and stored at −80 °C for further use. Total RNA was extracted from the pelleted cells with the RiboPure™ Bacteria kit (Ambion, USA) according to the manufacturer's instructions[30].

Before constructing the transcriptome library, the total RNA concentration and size distribution was determined on an Agilent 2100 Bioanalyzer (Agilent, USA). Messenger RNA (mRNA) was enriched by removing ribosomal RNA from total RNA with the MICROB*Express*™ Bacterial mRNA Enrichment Kit (Ambion, USA). Enriched mRNA was then fragmented and reverse transcription, adapter ligation, and amplification was performed afterwards using Ion Total RNA-Seq Kit (Ion Torrent, USA) according to the manufacturer's instructions. Qualities of the libraries were checked with the Agilent 2100 Bioanalyzer and the Agilent High Sensitivity DNA Kit (Agilent, USA). Libraries were equimolar pooled (20 pM) and fragments were amplified to Ion Sphere particles using the Ion One Touch™ 2 Instrument and Ion PGM™ Template OT2 200 Kit v2 (Life Technologies, USA) according to the manufacturer's instructions. After enrichment of the Template-Positive Ion Sphere™ Particles using the Ion One Touch™ ES (Life Technologies, USA), they were loaded on an Ion 318 v2 Chip. Subsequently, DNA fragments were sequenced according to the Ion PGM™ 200 Sequencing Kit using 125 cycles (500 flows). Each transcriptome analysis was performed in triplicate. The statistical significance of the triplicate sequencing for each library were examined by *T*-Test in R program[31] and showed no significant difference ($p > 0.05$). Thus, reads obtained from each independent sequencing run were combined before they were mapped to the reference genome.

Analyses of transcriptome reads were performed by the CLC Genomics Workbench software (Version 7.0.3, CLC Bio, Denmark). All reads were size-trimmed and quality-trimmed (maximum number of ambiguities: 2; Quality scores: 0.05) and then mapped to the *K. stuttgartiensis* genome (accession number PRJNA16685) using RNA-Seq analysis tool with a minimum length of 95% and a minimum identity of 95%. Before further analyses, transcriptome data of each library were normalized according to the expression value of two RNA polymerase genes rpoB and rpoC (kuste2957, kuste2958)[32]. The changes of gene expression level between NO supplied samples (Reactors II and III) and control sample (Reactor I) were identified by comparing the normalized RPKM (Reads Per Kilobase of exon model per Million mapped reads) value of all CDS to each other (Reactor II vs. Reactor I, Reactor III vs. Reactor I).

**Protein and proteome sample preparation and analysis**. On day 20 equal amount of biomass (50 ml) were harvested from all three reactors and pelleted by centrifugation for 20 min at $13,000 \times g$. Pelleted cells were resuspended in 3 ml $KH_2PO_4$ (20 mM, pH 7) solution and 2 ml $KH_2PO_4$, and EDTA-free protease inhibitor cocktail tablet (Roche, Switzerland) was added. Resuspended cells were then lysed with 8 M urea in 10 mM Tris–HCl (pH 8) solution for 30 min at room temperature and sonicated in a bath sonicator for 60 s. Cell remnants were pelleted by centrifugation for 20 min at $13,000 \times g$ at 4 °C. The supernatants containing protein were transferred to new tubes and stored at −80 °C until further analyses. Protein concentrations were measured using the Biuret method[33].

Samples were subjected to *in-solution* tryptic digestion as described elsewhere[34]. Briefly, proteins were reduced in 10 mM DTT for 30 min at room temperature prior to alkylation by 50 mM chloroacetamide. Proteins were pre-digested using LysC in a 1:50 LysC:protein ratio for 3 h at room temperature after which the sample was diluted 1:3 with 50 mM ammonium bicarbonate. Trypsin was added in a 1:50 trypsin:protein ratio for overnight digestion at 37 °C. After digestion, all samples were centrifuged to spin down all droplets and 2% trifluoroacetic acid was added 1:1 to the samples. Subsequent peptide mixtures were desalted and concentrated using C18 Omix tips (Agilent Technologies, USA).

Each sample was analyzed four times by C18 reversed phase liquid chromatography with online tandem mass spectrometry (LC–MS/MS). Measurements were performed using a nanoflow ultra-high-pressure liquid chromatograph (nano-Advance; Bruker Daltonics, USA) coupled online to an orthogonal quadrupole time-of-flight mass spectrometer (maXis 4G ETD; Bruker Daltonics, USA) via an electrospray ionization source (Captive sprayer; Bruker Daltonics, USA). Five microliters of tryptic digest were loaded onto the trapping column (Acclaim PepMap 100, 75 μm × 2 cm, nanoViper, 3 μm 100 Å C18 particles; Thermo Scientific, USA) using 0.1% formic acid at 7000 nl min$^{-1}$ for 3 min. Next, peptides were separated on a C18 reversed phase analytical column (Acclaim PepMap RSLC, 75 μm × 15 cm, nanoViper, 2 μm 100 Å C18 particles; Thermo Scientific, USA) at 40 °C using a linear gradient of 5–35% acetonitrile 0.1% formic acid in 60 min at 600 nl min$^{-1}$. The mass spectrometer was operated in positive ion mode to acquire line spectra in the mass range of 150–2200 $m/z$. Data-dependent acquisition of MS/MS spectra (AutoMSn) was performed using a 3 s duty cycle at 2 Hz acquisition rate for full MS spectra and a variable number of MS/MS experiments at precursor intensity scaled spectra rate (3 Hz MS/MS spectra rate @ 2000 counts, 20 Hz MS/MS spectra rate @ 100,000 counts). Precursor ions within the range of 400–1400 $m/z$ with chargestate $z \geq 2+$ were selected for MS/MS analysis with active exclusion enabled.

Protein identification and relative quantitation was performed using the MaxQuant software (v.1.6.2.10)[35] and the Andromeda database search algorithm. Extracted MS/MS spectra were searched against the NCBI RefSeq *K. stuttgartiensis* proteome database with added sequences of known contaminant proteins. The following settings were used for peptide and protein identification: carbamidomethyl (Cys) as fixed modification, oxidation (Met), and deamidation (NQ) as variable modifications, predefined MS and MS/MS settings for TOF instruments, minimal peptide length of six amino acids and a maximum allowed false discovery rate of 1% at both the peptide and protein level. Label-free quantitation (LFQ) was performed with the match between runs and re-quantify options using at least 2 razor + unique peptides. Retention time alignment was performed with a time alignment window of 20 min and a retention time match window of 0.5 min. LFQ values were used for subsequent data analysis. Proteins quantified in at least 3 out of 4 measurements for any growth condition were analyzed for differential expression using the analysis of variance (ANOVA) method with Bonferroni multiple testing correction. Proteins with adjusted *p*-value < 0.05 were considered to be significantly regulated.

**Reporting summary**. Further information on experimental design is available in the Nature Research Reporting Summary linked to this article.

## Data availability
The source data underlying Fig. 1a–c are provided as Source Data file. The transcriptome sequences from all three reactors have been deposited in the Sequence Read Archive (SRA) under accession number PRJNA485513. Proteomics data have been deposited to the ProteomeXchange Consortium via the PRIDE[36] partner repository with dataset identifier PXD011763.

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

## Acknowledgements

The authors thank Marcel Kuypers, Jan Keltjens, and Katharina Ettwig for critical reading of the manuscript. Z.H. and M.S.M.J. were supported by a European Research Council advanced grant (232937). M.S.M.J. was supported by a European Research Council advanced grant (339880). B.K. was supported by a European Research Council starting grant (640422).

## Author contributions

B.K. and Z.H. designed and analyzed experiments. Z.H. maintained cultures, performed activity experiments, analytical measurements and prepared material for transcriptomics and proteomics. Z.H. and T.v.A. performed metatranscriptomics analyses, H.J.C.T.W. performed metaproteomics analyses. Z.H., H.J.C.T.W., M.S.M.J., T.v.A. and B.K. discussed results. B.K. wrote the paper with input from all coauthors.

## Additional information

**Competing interests:** The authors declare no competing interests.

