## [Peer Review File · Nature Communications]

Reviewers' comments:

Reviewer #1 (Remarks to the Author):

In this paper, the authors show that the anammox bacterium *Kuenenia stuttgartiensis* can couple ammonium oxidation to the reduction of nitric oxide, when grown in the absence of nitrite. The sole end product of NO reduction is N₂. This study adds some interesting and novel details to the physiology of anammox, information that will be of broad interest and significance.

Specific comments

1. Line 25. "...its [NO's] direct use by microorganisms for growth was not demonstrated before". The implication is that it has not previously been reported that exogenously supplied NO can be a substrate for denitrification. However, at least one paper (PMID: 15583163) reporting NO-dependent growth seems to contradict the above statement. Even if it has not been shown before, it is not at all surprising that exogenously provided NO can function as an electron acceptor to support growth. In the opinion of this reviewer, the authors' claim to novelty (and impact) is overstated.
2. Lines 49-50. "...there are no known microorganisms that can use NO as the terminal electron acceptor". This statement is very confusing since (as the authors know well) the denitrifying bacteria surely use NO as a terminal electron acceptor, as is supported by a lot of old and new literature (one example, PMID 1317404 shows that electron transfer to NO from NADH generates a membrane potential). I am not sure what the authors are trying to say. Perhaps it is that growth with NO as the exogenous terminal electron acceptor has not been reported? But, see my comment above, and the reference.
3. Lines 51-52. Again, a confusing statement since many microorganisms with this capability are known, and some inferences can be made about the evolution of NO respiration.
4. Equations. Electrons should be shown with negative charges.
5. Figure 1. I am a bit confused about the units shown on the y-axes. According to the legend, the graphs show concentrations of ammonia and nitrite, but the unit shown on the axis is mmol, which is not a concentration. For NO, the legend refers to NO consumption, in which case the unit (mmol) does not seem correct. The text (line 102) reports an NO consumption rate in mmol/day, should the unit in the graphs be the same? In general, the authors need to check that there is consistency between the text, the figure and the figure legend.
6. Lines 98-99. The sentence refers to ammonium concentrations, but the units (mg-N/day and mmol/day) are of rates not concentrations, and are inconsistent with the figure.
7. Lines 129-130. Units in text are inconsistent with Figure. Should be mM on the y-axes?
8. Lines 14-143. "...when growing on NO-dependent ammonium oxidation..." does not make sense.
9. The experiments the authors describe are rather long-term (cultures grown for >50 days) and require somewhat specialized bioreactors and analytical methods. I recognize that this is technically demanding work, nevertheless it is the case that the paper apparently describes the results of a single experiment, in which three reactors were grown under different conditions. I think the authors should make some comment about reproducibility since it may be the case that independently cultured bioreactors do not necessarily follow the same trajectories.

Reviewer #2 (Remarks to the Author):

This is an important and clearly written paper with the finding that Anammox bacteria are capable of using nitrous oxide (NO) as the only electron acceptor for autotrophic growth on ammonium as the electron donor and energy source. It is an important complement to the early short paper of Kartal et al (ref 24) in AEM 2010.

The conclusion that this research has proven that *Kuenenia stuttgartiensis* can grow on NO is solid, but on a few detailed items the claims are somewhat oversold.

Figure 1 clearly shows the essential results obtained with highly enriched continuous membrane-cultures of *Kuenenia stuttgartiensis*, which is dominant in these cultures by at least 95%. Reproducibility of the results is intrinsic in the continuous mode of the culture which (unlike batch experiments) continues to consume NO over a long period of time. In the reactors the culture keep growing whilst part of the culture is removed by "bleeding". The culture of reactor II receiving both ammonium, nitrite and NO shows increase of biomass expected from the results of reactor III, which is an independent proof that these cultures are capable of using NO as electron acceptor. This apparently also applies to other anammox bacteria such as *Kuenenia stuttgartiensis* and *Brocadia fulgida* and *Brocadia anammoxidans* (see ref . 24)

Line 98. The test in ref 24 was *Kuenenia stuttgartiensis* and another remark in ref 24 refers to *Brocadia anammoxidans* (Schmidt et al 2002 in ref 24) The present paper is about a dominant culture of *Kuenenia stuttgartiensis* and this must be mentioned, one or another way.

Line 106-110. The conclusion in line 106-110 (repeated in concluding remarks line 193-195) is not correct with respect to the observed ratio of NO to oxidized ammonium, since the authors did not take into account the reducing equivalents ("electrons") required for CO₂ fixation. Indeed the biomass production is not given in equation (5). Given that ammonium is the electron donor a significant amount of this donor is required for biomass production. If we assume that the overall reaction of energy conversion plus biomass production is analogous to the reaction provided in Strous et al . (1999) (ref 25) then in the order of 15-20 % of the ammonium would be consumed for biomass production. This would mean that the observed ratio of 1.59 is not close to the "predicted stoichiometry" but rather too high. The author must explain why this is the case or at least not ignore this point. It is strongly recommended to include biomass data (OD is only an indication) in terms of protein, carbon and/or dry biomass.

Line 112-122 The conclusion in the paragraph concerning N₂O production line 112-122 needs a better explanation. Here the authors draw a conclusion on the basis of transcriptomics (no change in transcription of a potential detoxifying protein encoded by *norVW* (kuste3160), whilst a message in this paper on line 186-191 strongly emphasizes that slowly growing organisms like *Kuenenia stuttgartiensis* may not turnover their proteins very fast. If it is assumed that the anammox bacteria maintain the same NO detoxifying enzyme concentration (producing N₂O) the only expectation might be a somewhat increased N₂O production. If the (selected also slowly growing) contaminants are the culprits for N₂O production why then no more N₂O production if NO is increased?

In same paragraph (112-122) it may come as a surprise to the less well informed reader that the culture contains < 5% contaminants: nowhere in the paper it is mentioned that the authors are growing a highly enriched culture (in a membrane reactor) . This should at least be mentioned in the Materials and methods.

Line 148 English: to one another.

Line 163-172. The authors do not mention that kustc0458 and kustc045 are in Table S2. The same is true for kuste4574. It is recommended that these data are included in table 1, as they are not easily found in the supplementary tables. Removal of some other (irrelevant) data from Table 1 is recommended. If that is realized the paragraph will read more easily since on line 172 Table 1 is used again. Here the hypothesis is made that "kustc0458 could be one of the main enzymes responsible for nitrite reduction to NO in *K. stuttgartiensis*. Unfortunately this cautious hypothesis is overstated in the conclusions when it is stated in line 206 that the observations "singled out the HAO protein encoded by kustc0458 as the most likely enzyme that reduces nitrite to NO". It is suggested to just state that it is "the most likely candidate for the enzyme that"

Line 202 it should read: "by coupling nitrite reduction to NO with nitrite oxidation to nitrate"

Line 203 "both nitrite and NO disproportionation". The disproportionation of NO comes in as a surprise with no reference. Informed readers know that this is performed by anaerobic methane oxidizers (like *Methylomirabilis oxyfera*), but in this discussion it should either be left out or be explained properly with a reference.

Line 205. Change "Under these growth conditions" into "Under reactor III conditions with NO feed only"

Line 207: see remark 163-172. Add "candidate for the enzyme".

LineThe referee appreciates the speculation about the evolutionary sequences in the development of nitrite reduction to NO. If the evidence for the substantial production of NO indeed is strong the use of NO might have preceded the use of nitrite as substrate for NO production.

General: Will the authors generalize their finding to other Anammox Bacteria like *Brocadia* species being capable of metabolizing external NO?

Reviewer #3 (Remarks to the Author):

The authors have made impressive progress in understanding the bioenergetics and enzymology of the Anammox process over the past decade. This has allowed them to propose that *Kuenenia stuttgartiensis* can grow in the absence of nitrite (a source of endogenous NO) by coupling the condensation of NO and NH_4^+ to form hydrazine (N_2H_4) which consumes three electrons, to the subsequent oxidation N_2H_4 to dinitrogen (N_2) which yields four electrons. The extra electron is used directly for carbon assimilation to support growth.

In principle the study reported here that tests that hypothesis is both important and will be of interest to the broad readership of *Nature Communications* for two reasons: (1) it describes a mechanism through which a micro-organism can remove NO from the environment without formation of N_2O a potent greenhouse gas as an intermediate and (2) it provides insight into the bioenergetic strategies of pre-aerobic life on earth. Although when this is referred to in the manuscript (p3 line 50) it would be helpful to the reader to be given some information as to the geological timescales under discussion.

I have three difficulties with the manuscript as it stands, one conceptual and the others experimental/technical

The claim that "the direct use of NO by microorganisms for growth was not demonstrated before could be better articulated". Denitrifying bacteria use NO as a terminal electron acceptor for a protonmotive electron transfer chain which supports microbial growth and indeed one of the co-

authors has published a paper showing that *Nitrosomonas europae* can grow anaerobically using NO as a sole electron acceptor (Schmidt et al 2004 Microbiology 150: 4107-4114). Perhaps the authors could be more specific as to what they are claiming to be novel here. Also the authors should note that the role of free NO as an intermediate in denitrification was first reported by Carr et al EJB (1989) 179 683-92 reference 8 should be changed accordingly.

(2) Whilst I understand that continuous culture experiments of an organism as slow growing as *Kuenenia stuttgartiensis* are very long term, basing any study on a single biological replicate under each experimental condition inevitably calls into question the reproducibility of the study. Can the authors offer some justification for this approach.

(3) It would be much easier to review this manuscript if consistent dimensions were used throughout. Specifically:

The concentrations of bulk solutions of nutrients (p11) are given in mM but it would appear this is prior to mixing in a 1:1 ratio to form the influent. Was the [NH₄⁺] in bioreactor 3 really 15 mM or was it in fact 20 mM. How does 10000ppm NO translate into concentration in the gas and liquid phase (they are presumably not the same given limited solubility of NO in aqueous solutions).

The measured concentrations of key metabolites (p5) are given as an amount per day of N. It is not clear if this is a rate or if the "/day" indicates an individual data point. It is always useful in these types of experiment to see a mass balance reconciliation, but the underlying assumptions need to be placed in the supplementary materials.

The caption and Y axis of Figure 1 gives concentrations of analytes as amounts (relative to what) and don't discriminate between the gaseous and liquid phases – although the methods section suggests that NO is measured in the gas phase and all other species in the liquid phase. The caption should contain sufficient detail to allow interpretation by the reader.

I would be more than happy to consider a revised version of this manuscript that addresses these issues.

RESPONSES TO THE REFEREES:

We would like to thank the referees for the enthusiasm and interest they show for our work on nitric oxide dependent anaerobic ammonium oxidation. We are very pleased to see that the referees agree that our work will be interesting for the general scientific community. The interest of the referees is also apparent from their detailed comments, which will no doubt will make our manuscript clearer for the readers. We really appreciate their efforts.

Reviewer #1 (Remarks to the Author):

In this paper, the authors show that the anammox bacterium *Kuenenia stuttgartiensis* can couple ammonium oxidation to the reduction of nitric oxide, when grown in the absence of nitrite. The sole end product of NO reduction is N₂. This study adds some interesting and novel details to the physiology of anammox, information that will be of broad interest and significance.

Specific comments

1. Line 25. "...its [NO's] direct use by microorganisms for growth was not demonstrated before". The implication is that it has not previously been reported that exogenously supplied NO can be a substrate for denitrification. However, at least one paper (PMID: 15583163) reporting NO-dependent growth seems to contradict the above statement..

Indeed, we are aware of Schmidt et al., a paper where Dr. Jetten is also a coauthor. The Schmidt paper describes how *Nitrosomonas europaea* responds to additions of small amounts of nitric oxide in short-term incubations. At the time, it was not known that NO is an obligate intermediate of aerobic ammonium oxidation (Carranto et al. 2017). The experiments in the Schmidt show that *Nitrosomonas europaea* is "denitrifying" when fed with small amounts of NO as electron acceptor and H₂ as electron donor. There, the growth rate was very, very low (as the authors of that paper acknowledge), and was approximately 4% of the growth rate exhibited by aerobic ammonia-oxidation by *Nitrosomonas europaea*. Such a low growth rate is close to the detection limit of the methods employed in the Schmidt paper. The consensus in the nitrogen cycling field is that it is much more likely that the *Nitrosomonas* cells were exhibiting a stress response rather than growth. This is corroborated by follow up experiments in the Jetten labs other labs (DOI: 10.1016/j.syapm.2008.01.002. | DOI: 10.1080/09593330.2012.758666 | 10.1128/AEM.00668-12) as well as past unpublished experiments performed in the Jetten lab, which were not able to demonstrate appreciable NO-dependent growth.

Even if it has not been shown before, it is not at all surprising that exogenously provided NO can function as an electron acceptor to support growth. In the opinion of this reviewer, the authors' claim to novelty (and impact) is over-stated

Indeed, it is not surprising that "exogenously provided NO can function as an electron acceptor to support growth". But it is very surprising that no one else has shown it before. Already in our original manuscript we tried to bring this across with the following sentence: "Intriguingly, there are no known microorganisms that can use NO as the terminal electron acceptor."

2. Lines 49-50. "...there are no known microorganisms that can use NO as the terminal electron acceptor". This statement is very confusing since (as the authors know well) the denitrifying bacteria surely use NO as a terminal electron acceptor, as is supported by a lot of old and new literature (one example, PMID 1317404 shows that electron transfer to NO from NADH generates a membrane potential). I am not sure what the authors are trying to say. Perhaps it is that growth with NO as the exogenous terminal electron acceptor has not been reported? But, see my comment above, and the reference.

Indeed, the referee is right that we are aware NO is an electron acceptor in the electron transport chain of denitrifying microorganisms. It is correct that over the years, it was shown that small amounts of NO supplied together with artificial electron donors (e.g. NADH) are converted by denitrifying microorganisms. It has also been shown that many denitrifying organisms are inhibited by NO. Nevertheless, the question is still open whether there is a specialist denitrifier that can only grow on externally supplied NO in an efficient manner.

Here, we did mean activity and growth when NO was supplied as an exogenous terminal electron acceptor was never shown before. We rephrased the relevant parts of our manuscript to make sure that it is clear we mean external nitric oxide is used as an electron acceptor for the first time.

3. Lines 51-52. Again, a confusing statement since many microorganisms with this capability are known, and some inferences can be made about the evolution of NO respiration.

Here as well exogenous NO as an electron acceptor is meant. This sentence has also been changed. NO reductases that many organisms have and hydrazine synthase that anammox bacteria have are fundamentally different protein complexes. While NO reductases make an N-N bond through combining two NO molecules, hydrazine synthase combines NO with ammonium through a unique mechanism. We hope that the referee agrees understanding physiology of NO respiration through the anammox pathway does indeed help us understand the evolution of the nitrogen cycle. Furthermore, since ammonium was very abundant in early earth, it is conceivable that ammonium-dependent NO respiration evolved quite early in the evolutionary history of earth.

4. Equations. Electrons should be shown with negative charges.

This has been corrected.

5. Figure 1. I am a bit confused about the units shown on the y-axes. According to the legend, the graphs show concentrations of ammonia and nitrite, but the unit shown on the axis is mmol, which is not a concentration. For NO, the legend refers to NO consumption, in which case the unit (mmol) does not seem correct. The text (line 102) reports an NO consumption rate in mmol/day, should the unit in the graphs be the same? In general, the authors need to check that there is consistency between the text, the figure and the figure legend.

6. Lines 98-99. The sentence refers to ammonium concentrations, but the units (mg-N/day and mmol/day) are of rates not concentrations, and are inconsistent with the figure.

7. Lines 129-130. Units in text are inconsistent with Figure. Should be mM on the y-axes?

We are sorry to have created confusion and frustration concerning units displayed in our graphs and in the text. Thank you for pointing these discrepancies, as our results cannot be fully appreciated without displaying the correct units. In the revised manuscript concentrations are displayed as mg-N/L with corresponding mM values are reported in brackets to provide clear values to scientists that prefer either of these concentration units. In the instances where a rate is reported, the used unit is mg-N/L/d.

8. Lines 14-143. "...when growing on NO-dependent ammonium oxidation..." does not make sense.

We rephrased this sentence to make it clearer.

9. The experiments the authors describe are rather long-term (cultures grown for >50 days) and require somewhat specialized bioreactors and analytical methods. I recognize that this is technically demanding work, nevertheless it is the case that the paper apparently describes the results of a single experiment, in which three reactors were grown under different conditions. I think the authors should make some comment about reproducibility since it may be the case that independently cultured bioreactors do not necessarily follow the same trajectories.

We would like to point out that referee #2 states that "the reproducibility of the experiments is intrinsic in continuous cultivation experiments unlike batch incubations":

Figure 1 clearly shows the essential results obtained with highly enriched continuous membrane- cultures of *Kuenenia stuttgartiensis*, which is dominant in these cultures by at least 95%. Reproducibility of the results is intrinsic in the continuous mode of the culture which (unlike batch experiments) continues to consume NO over a long period of time. In the reactors the culture keep growing whilst part of the culture is removed by "bleeding". The culture of reactor II receiving both ammonium, nitrite and NO shows increase of biomass expected from the results of reactor III, which is an independent proof that these cultures are capable of using NO as electron acceptor." Furthermore, indeed as the referee #2 points out, in both reactor II and III NO-dependent ammonium oxidation is observed.

Compared to batch cultures, which require multiple replications, in continuous cultures, a similar effect can be created by biomass removal. Biomass removal forces cells to multiply at a specific rate, which is defined by the biomass washout rate. If the newly growing cells would not have the same growth rate, they would be washed out, if their activity would change, this would be immediately reflected in the measured substrates and end-products.

We now added a sentence to the beginning of the results section which clarifies this point.

Reviewer #2 (Remarks to the Author):

This is an important and clearly written paper with the finding that Anammox bacteria are capable of using nitrous oxide (NO) as the only electron acceptor for autotrophic growth on ammonium as the electron donor and energy source. It is an important complement to the early short paper of Kartal et al (ref 24) in AEM 2010.

The conclusion that this research has proven that *Kuenenia stuttgartiensis* can grow on NO is solid, but on a few detailed items the claims are somewhat oversold.

Figure 1 clearly shows the essential results obtained with highly enriched continuous membrane- cultures of *Kuenenia stuttgartiensis*, which is dominant in these cultures by at least 95%. Reproducibility of the results is intrinsic in the continuous mode of the culture which (unlike batch experiments) continues to consume NO over a long period of time. In the reactors the culture keep growing whilst part of the culture is removed by “bleeding”. The culture of reactor II receiving both ammonium, nitrite and and NO shows increase of biomass expected from the results of reactor III, which is an independent proof that these cultures are capable of using NO as electron acceptor. This apparently also applies to other anammox bacteria such as *Candidatus Brocadia fulgida* and *Candidatus Brocadia anammoxidans* (see ref . 24) We thank the referee on his/her comments about continuous cultivation as compared to batch incubations.

Line 98. The test in ref 24 was *Candidatus Brocadia fulgida* and another remark in ref 24 refers to *Brocadia anammoxidans* (Schmidt et al 2002 in ref 24) The present paper is about a dominant culture of *Kuenenia stuttgartiensis* and this must be mentioned, one or another way.

This is now included in the revised manuscript, and we generalized our findings to all anammox bacteria as suggested below.

Line 106-110. The conclusion in line 106-110 (repeated in concluding remarks line 193-195) is not correct with respect to the observed ratio of NO to oxidized ammonium, since the authors did not take into account the reducing equivalents (“electrons”) required for CO₂ fixation. Indeed the biomass production is not given in equation (5). Given that ammonium is the electron donor a significant amount of this donor is required for biomass production. If we assume that the overall reaction of energy conversion plus biomass production is analogous to the reaction provided in Strous et al . (1999) (ref 25) then in the order of 15-20 % of the ammonium would be consumed for biomass production. This would mean that the observed ratio of 1.59 is not close to the “predicted stoichiometry” but rather to high. The author must explain why this is the case or at least not ignore this point.

The reviewer’s comment is based on a misunderstanding. Anaerobic ammonium-oxidizing bacteria fix 0.066 mol of carbon into biomass per mol of ammonium they oxidize (displayed by equations in Strous et al 1999). Per 1 mol of this fixed carbon, they incorporate 0.15 mol N into the cells. This means that 0.99 % ($0.066 \cdot 0.15 \cdot 100$) of the supplied ammonium is incorporated into biomass as N.

It is strongly recommended to include biomass data (OD is only an indication) in terms of protein, carbon and/or dry biomass.

Optical density, protein or dry biomass are all indications and/or approximations of biomass amount. Only performing cell counts would be direct measurement of cell numbers. Here we had used OD as an internationally accepted way to represent a single-cell suspended culture. In the revised version of our manuscript, we changed the units of the second y axis to cell numbers per mL.

Line 112-122 The conclusion in the paragraph concerning N₂O production line 112-122 needs a better explanation. Here the authors draw a conclusion on the basis of transcriptomics (no change in transcription of a potential detoxifying protein encoded

by norVW (kuste3160), whilst a message in this paper on line 186-191 strongly emphasizes that slowly growing organisms like Kuenenia species may not turnover their proteins very fast. If it is assumed that the anammox bacteria maintain the same NO detoxifying enzyme concentration (producing N₂O) the only expectation might be a somewhat increased N₂O production. If the (selected also slowly growing) contaminants are the culprits for N₂O production why then no more N₂O production if NO is increased?

Here, we can only speculate as to what the side population might be doing. It could be that anammox bacteria are better converters of NO. But the more likely explanation is that even though the NO in the influent stream is increased there is no external electron donor that these organisms might be using is added. In other words, the “contaminant denitrifiers” are still electron donor limited as the only electron donor supplied is ammonium, and only anammox bacteria can grow on ammonium under anaerobic conditions. If there would be more electron donor supplied, the “contaminant denitrifiers” would be able to convert more NO to N₂O. We have now rephrased this sentence to indicate that the other community members most likely use organic carbon from cell decay.

In same paragraph (112-122) it may come as a surprise to the less well informed reader that the culture contains < 5% contaminants: nowhere in the paper it is mentioned that the authors are growing a highly enriched culture (in a membrane reactor) . This should at least be mentioned in the Materials and methods.

We thank the referee for pointing this out. We now mentioned that this is a highly enriched culture in materials and methods and in results.

Line 148 English: to one another.

Rephrased.

Line 163-172. The authors do not mention that kustc0458 and kustc045 are in Table S2. The same is true for kuste4574. It is recommended that these data are included in table 1, as they are not easily found in the supplementary tables. Removal of some other (irrelevant) data from Table 1 is recommended. If that is realized the paragraph will read more easily since on line 172 Table 1 is used again. Here the hypothesis is made that “kustc0458 could be one of the main enzymes responsible for nitrite reduction to NO in K. stuttgartiensis. Unfortunately this cautious hypothesis is overstated in the conclusions when it is stated in line206 that the observations “singled out the HAO protein encoded by kustc0458 as the most likely enzyme that reduces nitrite to NO”. It is suggested to just state that it is “the most likely candidate for the enzyme that”

Rephrased as suggested. We added these enzymes to table 1, but we did not remove any that were already in there as we believe that all displayed enzymes are relevant as well as kuste4574.

Line 202 it should read: “by coupling nitrite reduction to NO with nitrite oxidation to nitrate”

Rephrased as suggested. We also included a new equation (Equation 6) to make it clear what is meant here.

Line 203 “both nitrite and NO disproportionation”. The disproportionation of NO comes in as a surprise with no reference. Informed readers know that this is performed by anaerobic methane oxidizers (like *Methylomirabilis oxyfera*), but in this discussion it should either be left out or be explained properly with a reference. The reaction performed by *Methylomirabilis oxyfera* and related species is NO dismutation into N₂ and O₂. *M. oxyfera* does not disproportionate nitrite. Here, we are discussing real disproportionation of nitrite as it is reduced to NO and is at the same time being oxidized to nitrate. This reaction is included now as equation 6. In the conclusion paragraph, we are not referring to NO or nitrite conversion into N₂ and O₂, but again we are discussing true disproportionation. These reactions are depicted by following five equations:

We believe that including these reactions in the conclusions would distract attention of the reader from the main point of the paper. Therefore, instead of the complete balanced equations, we spelled out the possible combinations in brackets in text.

Line 205. Change “Under these growth conditions” into “Under reactor III conditions with NO feed only”
Rephrased as suggested.

Line 207: see remark 163-172. Add “candidate for the enzyme”.
Rephrased as suggested

LineThe referee appreciates the speculation about the evolutionary sequences in the development of nitrite reduction to NO. If the evidence for the substantial production of NO indeed is strong the use of NO might have preceded the use of nitrite as substrate for NO production.

We thank the referee for sharing our excitement about the possible impact of our work on the evolution of the nitrogen cycle. The fact that NO preceded other oxidized N forms is the general consensus of studies that speculate on the nature of N-oxides on early earth.

General: Will the authors generalize their finding to other Anammox Bacteria like *Brocadia* species being capable of metabolizing external NO?

We believe that this is true for all anammox species, and in the concluding remarks, we were trying to generalize to all anammox bacteria.

Reviewer #3 (Remarks to the Author):

The authors have made impressive progress in understanding the bioenergetics and enzymology of the Anammox process over the past decade. This has allowed them to propose that *Kuenenia stuttgartiensis* can grow in the absence of nitrite (a source of endogenous NO) by coupling the condensation of NO and NH₄⁺ to form hydrazine (N₂H₄) which consumes three electrons, to the subsequent oxidation N₂H₄ to dinitrogen (N₂) which yields four electrons. The extra electron is used directly for carbon assimilation to support growth.

In principle the study reported here that tests that hypothesis is both important and will be of interest to the broad readership of Nature Communications for two reasons: (1) it describes a mechanism through which a micro-organism can remove NO from the environment without formation of N₂O a potent greenhouse gas as an intermediate and (2) it provides insight into the bioenergetic strategies of pre-aerobic life on earth. Although when this is referred to in the manuscript (p3 line 50) it would be helpful to the reader to be given some information as to the geological timescales under discussion.

I have three difficulties with the manuscript as it stands, one conceptual and the others experimental/technical

The claim that “the direct use of NO by microorganisms for growth was not demonstrated before could be better articulated”. Denitrifying bacteria use NO as a terminal electron acceptor for a protonmotive electron transfer chain which supports microbial growth and indeed one of the co-authors has published a paper showing that *Nitrosomonas europae* can grow anaerobically using NO as a sole electron acceptor (Schmidt et al 2004 Microbiology 150: 4107-4114). Perhaps the authors could be more specific as to what they are claiming to be novel here.

Please see our response to referee #1 concerning this comment.

Also the authors should note that the role of free NO as an intermediate in denitrification was first reported by Carr et al EJB (1989) 179 683-92 reference 8 should be changed accordingly.

NO was suggested as an intermediate already in 1950s as a possible intermediate in the denitrification pathway (see for example PMID:13295215). NO was eventually recognized by all as an intermediate with purification and characterization of NO reductases. In order to give credits to all involved on NO related research over the years, we chose to cite the comprehensive review of Zumft et al.

(2) Whilst I understand that continuous culture experiments of an organism as slow growing as *Kuenenia stuttgartiensis* are very long term, basing any study on a single biological replicate under each experimental condition inevitably calls into question the reproducibility of the study. Can the authors offer some justification for this approach.

Please see our response to referee #1 for this point.

(3) It would be much easier to review this manuscript if consistent dimensions were used throughout. Specifically:

The concentrations of bulk solutions of nutrients (p11) are given in mM but it would appear this is prior to mixing in a 1:1 ratio to form the influent. Was the $[\text{NH}_4^+]$ in bioreactor 3 really 15 mM or was it in fact 20 mM. How does 10000ppm NO translate into concentration in the gas and liquid phase (they are presumably not the same given limited solubility of NO in aqueous solutions).

The measured concentrations of key metabolites (p5) are given as an amount per day of N. It is not clear if this is a rate or if the “/day” indicates an individual data point. It is always useful in these types of experiment to see a mass balance reconciliation, but the underlying assumptions need to be placed in the supplementary materials.

The caption and Y axis of Figure 1 gives concentrations of analytes as amounts (relative to what) and don't discriminate between the gaseous and liquid phases – although the methods section suggests that NO is measured in the gas phase and all other species in the liquid phase. The caption should contain sufficient detail to allow interpretation by the reader.

We are sorry to have created confusion and frustration concerning units displayed in our graphs and in the text. Thank you for pointing these discrepancies, as our results cannot be fully appreciated without displaying the correct units. In the revised manuscript concentrations are displayed as mg-N/L with corresponding mM values are reported in brackets to provide clear values to scientists that prefer either of these concentration units. In the instances where a rate is reported, the used unit is mg-N/L/d.

Concerning the medium feed, we did not mix media with NO. NO is bubbled independently through the bioreactor, and is not supplied in solution in the liquid phase. This is now explicitly stated in the materials and methods section.

I would be more than happy to consider a revised version of this manuscript that addresses these issues.

Thank you!

REVIEWERS' COMMENTS:

Reviewer #1 (Remarks to the Author):

This manuscript is much improved, and most of my concerns have been addressed by the authors. I have just the following residual comments.

1. Lines 48-49. "...there are no known microorganisms that can use external NO as the terminal electron acceptor". I remain uncomfortable with this sentence for the following reasons. Firstly, read at face value I do not think that this statement is correct. For example, PMID: 1317404 shows that NADH and succinate can be oxidized by externally supplied NO, with the accompanying generation of a membrane potential. That is to say, 'external' NO is used as the terminal electron acceptor (given everything else that is known, to argue otherwise would be perverse). Secondly, what I believe the authors mean to say is : "...there are no known microorganisms that can grow using external NO as the terminal electron acceptor". But, there surely are denitrifying bacteria that can grow on exogenous NO; I don't find it so surprising that nobody has bothered to confirm this experimentally, nor do I find the authors' statement to be intriguing. I suggest an alternative wording, something like this: "Robust microbial growth with externally supplied NO as the terminal electron acceptor has not previously been observed experimentally".

2. Line 100. Is this the calculated NO concentration in the liquid phase? If so, it is way above the solubility limit of NO in aqueous solution, so I wonder if the number is meaningful. Perhaps some clarification is needed?

3. Figure 1. NO concentrations are shown in panels b and c (open diamonds), but NO is not included in the y-axis label.

Referee 2 Line 106-110. The conclusion in line 106-110 (repeated in concluding remarks line

193-195) is not correct with respect to the observed ratio of NO to oxidized ammonium, since the authors did not take into account the reducing equivalents ("electrons") required for CO₂ fixation. Indeed the biomass production is not given in equation (5). Given that ammonium is the electron donor a significant amount of this donor is required for biomass production. If we assume that the overall reaction of energy conversion plus biomass production is analogous to the reaction provided in Strous et al. (1999) (ref 25) then in the order of 15-20 % of the ammonium would be consumed for biomass production. This would mean that the observed ratio of 1.59 is not close to the "predicted stoichiometry" but rather too high. The author must explain why this is the case or at least not ignore this point.

The reviewer's comment is based on a misunderstanding. Anaerobic ammonium-oxidizing bacteria fix 0.066 mol of carbon into biomass per mol of ammonium they oxidize (displayed by equations in Strous et al 1999). Per 1 mol of this fixed carbon, they incorporate 0.15 mol N into the cells. This means that 0.99 % ($0.066 \cdot 0.15 \cdot 100$) of the supplied ammonium is incorporated into biomass as N.

Referee 2: There is no misunderstanding. Indeed the contribution of NH₄⁺ to biomass as nitrogen source is small. As pointed out by the authors part of the electrons coming from hydrazine will be used for CO₂ fixation. Hence less NO is needed as electron acceptor. The other way around, if ammonium is the electron donor for CO₂ fixation, more NH₄⁺ must be converted to N₂ than the stoichiometric quantity to reduce NO, because CO₂ also must be reduced. In other words the total electron balance of the reaction including CO₂ fixation (reduction to biomass, see Strous et al) requires extra electrons from NH₄⁺. If about 20% of the ammonium is required for CO₂ fixation, the ratio NO/NH₄⁺ will be lower than 1.5. Instead of 3/2 you get $3/2.4 = 1.25$. If the yield in this case, with NO as acceptor, is much lower than in the case of nitrite as electron acceptor, the ratio would go up. This is why the referee wanted biomass carbon instead of Opt Density. Dry weight is usually 45-50 % C and this would allow calculation of the (approximate) amount of CO₂ assimilated.

Reviewer #1 (Remarks to the Author):

This manuscript is much improved, and most of my concerns have been addressed by the authors. I have just the following residual comments.

1. Lines 48-49. "...there are no known microorganisms that can use external NO as the terminal electron acceptor". I remain uncomfortable with this sentence for the following reasons. Firstly, read at face value I do not think that this statement is correct. For example, PMID: 1317404 shows that NADH and succinate can be oxidized by externally supplied NO, with the accompanying generation of a membrane potential. That is to say, 'external' NO is used as the terminal electron acceptor (given everything else that is known, to argue otherwise would be perverse). Secondly, what I believe the authors mean to say is : "...there are no known microorganisms that can grow using external NO as the terminal electron acceptor". But, there surely are denitrifying bacteria that can grow on exogenous NO; I don't find it so surprising that nobody has bothered to confirm this experimentally, nor do I find the authors' statement to be intriguing. I suggest an alternative wording, something like this: "Robust microbial growth with externally supplied NO as the terminal electron acceptor has not previously been observed experimentally".

There are several papers in literature that show NO consumption under artificial conditions, using artificial electron acceptors, in the presence and absence of inhibitors. However, there is no study showing continuous growth of microorganisms on nitric oxide reduction coupled to substrate oxidation in long-term incubations. To accommodate the suggestion of the referee, we have now rephrased the sentence at line 67-68. We genuinely find it intriguing that NO-dependent growth has not been shown before. Furthermore, our results show that anammox bacteria do not couple nitrite reduction to nitrate to cell carbon fixation, which fundamentally change our understanding of how anammox metabolism functions.

2. Line 100. Is this the calculated NO concentration in the liquid phase? If so, it is way above the solubility limit of NO in aqueous solution, so I wonder if the number is meaningful. Perhaps some clarification is needed?

This value is the concentration of nitric oxide converted by the microorganisms. It is calculated from the difference between the NO concentration in the influent gas phase and the NO concentration in the effluent of the gas phase taking the flow rate of the gas flow through the reactor. We have now added a sentence to the methods section on how this value was calculated.

3. Figure 1. NO concentrations are shown in panels b and c (open diamonds), but NO is not included in the y-axis label.

We thank the reviewer for pointing this out. It has been corrected.

Reviewer #2 (Remarks to the Author):

Referee 2 Line 106-110. The conclusion in line 106-110 (repeated in concluding remarks line 193-195) is not correct with respect to the observed ratio of NO to oxidized ammonium, since the authors did not take into account the reducing equivalents ("electrons") required for CO₂ fixation. Indeed the biomass production is not given in equation (5). Given that ammonium is the electron donor a significant amount of this donor is required for biomass production. If we assume that the overall reaction of energy conversion plus biomass production is analogous to the reaction provided in Strous et al. (1999) (ref 25) then in the order of 15-20 % of the ammonium would be consumed for biomass production. This would mean that the observed ratio of 1.59 is not close to the "predicted stoichiometry" but rather too high. The author must explain why this is the case or at least not ignore this point.

The reviewer's comment is based on a misunderstanding. Anaerobic ammonium-oxidizing bacteria fix 0.066 mol of carbon into biomass per mol of ammonium they oxidize (displayed by equations in Strous et al 1999). Per 1 mol of this fixed carbon, they incorporate 0.15 mol N into the cells. This means that 0.99 % ($0.066 \cdot 0.15 \cdot 100$) of the supplied ammonium is incorporated into biomass as N.

Referee 2:

There is no misunderstanding. Indeed the contribution of NH₄⁺ to biomass as nitrogen source is small. As pointed out by the authors part of the electrons coming from hydrazine will be used for CO₂ fixation. Hence less NO is needed as electron acceptor. The other way around, if ammonium is the electron donor for CO₂ fixation, more NH₄⁺ must be converted to N₂ than the stoichiometric quantity to reduce NO, because CO₂ also must be reduced. In other words the total electron balance of the reaction including CO₂ fixation (reduction to biomass, see Strous et al) requires extra electrons from NH₄⁺. If about 20% of the ammonium is required for CO₂ fixation, the ratio NO/NH₄⁺ will be lower than 1.5. Instead of $3/2$ you get $3/2.4 = 1.25$. If the yield in this case, with NO as acceptor, is much lower than in the case of nitrite as electron acceptor, the ratio would go up. This is why the referee wanted biomass carbon instead of Opt Density. Dry weight is usually 45-50 % C and this would allow calculation of the (approximate) amount of CO₂ assimilated.

We do not understand where the referee derives the assumption that 20% more ammonium needs to be oxidized for CO₂ fixation into biomass. In any case, the Strous et al. 1998 yield values are obtained from another anammox species (*Brocadia anammoxidans*), and describe a fundamentally different bioreactor system where nitrite is oxidized to nitrate for cell carbon fixation, which is not the case here; in fact, there is no nitrate production when the cells are fed with NO and ammonium. In order to determine a yield for growth of *Kuenenia stuttgartiensis* under NO-dependent ammonium oxidation conditions, a detailed mass balance would be required. This cannot be calculated from dry weight with sufficient confidence, because as the referee points out, carbon content from dry weight measurements would be an approximation. All in all, we believe detailed yield calculations and mass balances are beyond the scope of this manuscript.